# Clinical Spectrum of Schistosomiasis: An Update

**DOI:** 10.3390/jcm10235521

**Published:** 2021-11-25

**Authors:** Cristina Carbonell, Beatriz Rodríguez-Alonso, Amparo López-Bernús, Hugo Almeida, Inmaculada Galindo-Pérez, Virginia Velasco-Tirado, Miguel Marcos, Javier Pardo-Lledías, Moncef Belhassen-García

**Affiliations:** 1Servicio de Medicina Interna, Unidad de Infecciosas, Complejo Asistencial Universitario de Salamanca (CAUSA), Centro de Investigación de Enfermedades Tropicales de la Universidad de Salamanca (CIETUS), Instituto de Investigación Biomédica de Salamanca (IBSAL), 37007 Salamanca, Spain; carbonell@usal.es (C.C.); beamedicina@hotmail.com (B.R.-A.); alopezb@saludcastillayleon.es (A.L.-B.); mmarcos@usal.es (M.M.); 2Internal Medicine, Unidade Local de Saúde de Guarda, 6300-749 Guarda, Portugal; hugoalmeida6@gmail.com; 3Centro de Atención Primaria, 39740 Santoña, Spain; igalindop@hotmail.com; 4Servicio de Dermatología, CAUSA, CIETUS, IBSAL, 37007 Salamanca, Spain; virvela@yahoo.es; 5Servicio de Medicina Interna, Hospital Marqués de Valdecilla, Universidad de Cantabria, IDIVAL (Instituto de Investigación Valdecilla), 39008 Santander, Spain; javipard2@hotmail.com

**Keywords:** schistosomiasis, imported diseases, acute schistosomiasis, Katayama syndrome, cercarial dermatitis, chronic schistosomiasis, clinical manifestations

## Abstract

Schistosomiasis is a helminthic infection and one of the neglected tropical diseases (NTDs). It is caused by blood flukes of the genus Schistosoma. It is an important public health problem, particularly in poverty-stricken areas, especially those within the tropics and subtropics. It is estimated that at least 236 million people worldwide are infected, 90% of them in sub-Saharan Africa, and that this disease causes approximately 300,000 deaths annually. The clinical manifestations are varied and affect practically all organs. There are substantial differences in the clinical presentation, depending on the phase and clinical form of schistosomiasis in which it occurs. Schistosomiasis can remain undiagnosed for a long period of time, with secondary clinical lesion. Here, we review the clinical profile of schistosomiasis. This information may aid in the development of more efficacious treatments and improved disease prognosis.

## 1. Introduction

Schistosomiasis is a neglected tropical disease (NTD) and one of the most prevalent parasitic diseases worldwide, affecting more than 236 million people globally, according to data provided by the World Health Organization [1]. The three major schistosomes infecting humans are *Schistosoma mansoni*, *Schistosoma haematobium*, and *Schistosoma japonicum*, with less epidemiological impact from *Schistosoma intercalatum* and *Schistosoma mekongi* [1]. The importance of schistosomiasis for public health has increased exponentially in recent years due to the increase in migration and international tourism [1]. From a clinical point of view, schistosomiasis is divided into three stages: (i) the first occurs 24 h after the penetration of the cercariae into the dermis called cercarial dermatitis, (ii) acute schistosomiasis appears 3–8 weeks after infection, and (iii) the chronic stage occurs months or years after infection, and is a consequence of the formation of granulomas in the tissues around the schistosome eggs.

## 2. Cercarial Dermatitis or Swimmer’s Itch

Cercarial dermatitis or Swimmer’s itch is reported in 5–100% of exposed travelers; it develops a few hours after freshwater contact carrying infective cercariae, and the affected person develops an itchy maculo-papular rash, limited to areas immersed in water [2].

Several species are known to cause cercarial dermatitis; the most commonly implicated genus globally is the waterfowl schistosome *Trichobilharzia* spp. (*T. ocellata*, *T. brevis*, *T. stagnicolae*, *T. physellae*, *T. regenti*, and others). Other avian schistosomes that cause cercarial dermatitis include *Ornithobilharzia* spp., *Austrobilharzia* spp., *Bilharziella polonica*, and *Gigantobilharzia huronensis*. Cases involving mammalian schistosomes *Heterobilharzia americana*, *Bivitellobiharzia* spp., *Schistosomatium* spp., and some aberrant zoonotic *Schistosoma* spp. (*S. spindale*) occur occasionally. These schistosomes all use different snail intermediate hosts, commonly those from the families *Nassariidae*, *Lymnaeidae*, and *Physidae.*

The itch becomes more intense, and the rash typically develops with papules and vesicles within hours or a few days after exposure. The disease is self-limiting within 1–3 weeks; given the fleeting nature of the condition, itching is sometimes the only finding that patients present [3,4]. In this acute phase of infection, diagnosis is very difficult [5] despite recent advances in diagnostic methods, such as polymerase chain reaction (PCR) or antigen testing [6,7]. The history of dermatitis is very indicative of acute schistosomiasis, although other entities can produce it. Initially, serology can still be negative, and no eggs are found in urine or stool. In cases acquired on short trips, serology was the technique that most frequently led to diagnosis, indicating a low parasite load [8]. Within this general picture, four presentations have been differentiated, and their characteristics are summarized in Table 1.

Another clinical picture is a late onset that manifests as urticaria or angioedema after 1–12 weeks of exposure. In these cases, the lesions are erythematous, raised, with a size between 1 and 3 cm, and often accompanied by itching [9]. The time sequence regarding exposure to water is essential to guide the diagnosis correctly.

## 3. Acute Schistosomiasis

Acute schistosomiasis is a typically self-limited process seen primarily in nonimmune patients (travelers) (Table 2) and is due to a cercariae-induced hypersensitivity reaction. Diagnostic delay is frequent in nonendemic countries with subsequent clinical damage [10,11,12] and is basically divided into two clinical pictures: (i) Swimmer’s itch or cercarial dermatitis and (ii) Katayama fever or syndrome (Figure 1). The risk of infection increases with the duration and amount of exposures [3], and cercarial penetration episodes have been described after contact for only 1–5 min [4]. Assessing the history of exposure to fresh water (rivers or lakes) is essential to guide the diagnosis of acute schistosomiasis, so asking about activities, such as bathing, crossing rivers/lakes, or even showering, is essential [13].

Estimating the prevalence of acute schistosomiasis is complicated, because in many cases, the registries do not distinguish between acute and chronic forms [14]. In travelers, few studies have evaluated the presence of schistosomiasis [1,8,10,15,16,17,18,19,20]. Thus, different studies show that more than 3% of international travelers have schistosomiasis [16], exceeding 5% of diagnoses if travelers came from sub-Saharan Africa [21]. Gautret et al., described the imported infections of 7408 travelers, finding 152 schistosomiasis cases [1]. The highest proportion of schistosomiasis (40%) occurred in missionaries, volunteers, and humanitarian workers, followed by tourists (19%), visiting friends and relatives (16%), and immigrants (13%). Most of these patients returned from Africa, mainly from Egypt, Ghana, Malawi, Mali and Uganda [1]. A study by the European Network for Tropical Medicine and Travel Health (TropNet, http://www.tropnet.eu, accessed on 15 November 2021), which analyzed 1465 imported schistosomiasis cases from 1997 to 2010, noted that 95% of schistosomiasis cases in travelers were acquired in sub-Saharan Africa, with *S. haematobium* being the most frequent species [8].

Katayama fever or syndrome was first described in 1847 in Japan (Katayama district). This syndrome is usually produced by *S. japonicum* [9]. Katayama fever is currently not used because not all patients present with fever, and most cases are caused by *S. mansoni* or *S. haematobium*. Symptoms usually appear between 2 and 6 weeks after exposure during the maturation of adult forms [9]. However, incubation periods of 1 to 12 weeks have been described [22]. In this phase, the schistosomula matures until reaching the adult form, leading to mating and laying the eggs. It has been postulated that the cause of the condition is the passage of soluble antigens from the eggs to the blood, giving rise to an inflammatory response that can be more or less severe, depending on the species, although for other authors, it occurs even before laying the eggs. In nonimmune travelers, the infection is symptomatic in 54–100% of cases [5]. The most frequent symptoms are high fever, asthenia, myalgia, urticaria, and a nonproductive cough. Most patients recover spontaneously after 2–10 weeks, although some develop more severe and persistent disease, with weight loss, diarrhea, diffuse abdominal pain, hepatomegaly, dyspnea, and generalized skin rash. About half of the cases have eosinophilia [5]. Eosinophilia usually appears late, 21 days after fever and up to 47 days (range 25–119) after contact, so its absence does not exclude the diagnosis of acute schistosomiasis and requires repeat laboratory tests in 2–3 weeks [23]. The picture may be accompanied by anemia, elevated IgE, altered transaminases, hypoalbuminemia, and hypergammaglobulinemia.

In some cases, complications appear at the neurological, cardiac, and pulmonary levels that worsen the prognosis. Neurological symptoms develop in approximately 2% of acute schistosomiasis cases [24]. They usually appear three weeks after the systemic symptoms and among the most frequent manifestations we cite: (i) headache (usually transient or intermittent), (ii) altered consciousness (up to coma), (iii) seizures, (iv) aphasia, (v) blurred vision, (vi) cerebellar symptoms, and (vii) a picture of acute encephalopathy (frequently caused by *S. japonicum* and *S. mansoni*, and underdiagnosed) [24].

Additionally, cases of spinal cord involvement have been described, usually due to *S. mansoni* [25], which is usually more frequent in young men, although cases in the extreme ages of life have also been described. It usually presents as an acute or subacute myelopathic syndrome and/or polyradiculitis. Involvement in the T11-L1 segment is of particular importance, although the most frequent is involvement at the T6 level or lower, with cases reported at higher levels [25]. The appearance of the lesions on the Magnetic Resonance Imaging (MRI) is indistinguishable from other causes of transverse myelitis.

In addition, multiple cerebral infarcts have been described in the border area, as well as cerebral vasculitis during the first three months after infection [24], which may occur as a result of eosinophil-mediated toxicity [26]. Cerebral vasculitis associated with schistosomiasis in the absence of eosinophilia has also been described [27]. Imaging tests usually show cerebral edema and small contrast-enhancing multifocal lesions, usually located in the frontal, parietal, occipital, and brainstem lobes [24]. Cerebrospinal fluid may be normal or show nonspecific findings.

At the cardiac level, pictures of myocarditis, pericarditis or asymptomatic ischemia have been described [28]. In a study of 315 patients with acute schistosomiasis, the electrocardiogram showed T wave abnormalities in almost all patients (99%) and ST segment abnormalities in 52% of patients [29]. In another study with 31 patients infected with *S. mansoni*, it was observed that approximately 40% of the patients had chest pain and 20% had a diagnosis of pericarditis [30].

Acute pulmonary involvement usually appears 3–8 weeks after penetration of the schistosome, and its incidence is unknown. The most frequent clinical manifestations are dyspnea, bronchospasm, productive cough, dry cough, hemoptysis, chest pain, and wheezing [20]; in exceptional cases, pulmonary schistosomiasis can mimic IgG4-related lung disease [31]. Although they can coincide with the fever of Katayama syndrome, these symptoms can be independent of this fever and appear several weeks after the fever or continue after defervescence. In several series of travelers with Katayama syndrome, the prevalence of cough was above 40% [9,30]. The radiological involvement of acute pulmonary schistosomiasis is highly variable, ranging between radiological normality in highly symptomatic patients and florid radiological pictures in asymptomatic patients [32]. The main radiological findings are poorly defined pulmonary nodules or interstitial pneumonitis similar to that seen in tropical pulmonary eosinophilia.

## 4. Chronic Schistosomiasis

Chronic schistosomiasis occurs mainly as a result of granulomatous inflammation induced by the accumulation of eggs released by the parasite in the different tissues, given their ability to produce inflammation and fibrosis [11]. The eggs reach the different tissues after being transported through the portal venous system and embolizing in the liver or spleen or, in the case of passing to the systemic circulation, the lungs, the brain or the spinal cord. Eggs trapped in tissues secrete proteins and carbohydrates that induce a host Th-2 immune response, leading to an eosinophilic granulomatous reaction [12]. In this way, patients born in endemic countries acquire the infection during childhood and may develop a chronic disease because of continuous reinfection through freshwater contact, with slight clinical variations according to the schistosome species. Thus, sub-Saharan African countries account for 90% of schistosomiasis cases globally [33]. Prevalence rates range from 10 to 50% for *S. haematobium* in countries in s-Saharan Africa and the Middle East [34] and 1 to 40% for *S. mansoni* in sub-Saharan Africa and South America, with similar figures for *S. japonicum* in Indonesia, parts of China, and Southeast Asia [35]. A recent meta-analysis estimated a seroprevalence in immigrants from endemic areas and living in nonendemic areas at approximately 18%, with immigrants from sub-Saharan Africa having the highest prevalence (approximately 24%) [36]. The group of HIV + immigrants showed figures over 20% [37], and that of unaccompanied immigrant minors showed figures of approximately 16% [38]. In addition, in the immigrant group from endemic areas, approximately 40% of diagnosed infections occur in asymptomatic patients [8], which can cause a diagnostic delay [39] and sometimes has serious consequences [40].

Next, a clinical review will be presented according to organ and system involvement.

## 5. Intestinal Schistosomiasis

Intestinal schistosomiasis is a very frequent chronic complication that has been described in cases for more than 20 years [41] and is caused by infection with *S. mansoni*, *S. japonicum*, *S. intercalatum*, *S. mekongi*, and occasionally, *S. haematobium*. Its severity is related to recurrent exposure and the number of parasite eggs present in the mucosa of the intestinal tract, mainly the colon and rectum, where they cause an inflammatory reaction and the appearance of granulomas, ulcers, and fibrosis [8,15] (Figure 2 and Figure 3).

The most common symptoms are nonspecific and include chronic or intermittent abdominal pain, asthenia, weight loss, anorexia, and diarrhea [16,17]. In severe cases, it may be accompanied by anemia, secondary to bleeding from ulcerations in the colon and rectum (which can mimic chronic colitis of other etiologies, particularly inflammatory bowel disease) [18]. Additionally, patients have a higher incidence of colorectal polyps, mainly rectal polyps [42], and granulomatous inflammation can degenerate into polyps, which is the most common intestinal lesion in chronic intestinal schistosomiasis and even triggers the appearance of dysplasia; however, these polyps were all discovered during colonoscopy and presented as large polyps rather than cecal thickening [43]. In rare cases, they can even cause intestinal obstructive symptoms [19] mimicking colon cancer. Moreover, endoscopic imaging, and macroscopic manifestations are not specific for chronic intestinal schistosomiasis.

Some authors note that using a single urine sample could be very sensitive and highly specific in the diagnosis of intestinal schistosomiasis, using either the trace negative model of point of contact assay or conventional PCR, when compared with fecal egg detection using duplicate Kato-Katz fecal egg detection [44]. However, using a single tool restricts the management of the disease in areas of low endemicity [44]. Correct diagnosis relies on increased awareness of this disease and rigorous search for parasitic eggs in tissue, particularly in patients from endemic areas who are suspected to have inflammatory bowel disease.

## 6. Hepatosplenic Schistosomiasis

Hepatosplenic schistosomiasis is a heterogeneous condition, ranging from a mildly symptomatic to life-threatening disease [45] (Figure 4). The most common presentation of hepatosplenic schistosomiasis was upper gastrointestinal bleeding leading to severe anemia [46]. In contrast to cirrhosis, in hepatosplenic schistosomiasis, hepatic function is preserved overall [45]. Several studies have demonstrated a correlation between disease severity (measured by i.e., organomegaly) and quantitative fecal egg output [47].

Hematochemical alterations, such as thrombocytopenia and leukopenia, classically attributed to “hypersplenism,” might be due to intrasplenic blood stasis rather than sequestration with abnormal splenic function. Severe anemia may imply delayed presentation at the health facility and chronic blood loss due to prolonged bleeding or nutritional deficiency. An association of HBV or HCV coinfection with severe schistosomiasis has been documented in several studies, and coinfection with hepatitis viruses may accelerate the development of periportal fibrosis and related complications [46]. Two situations can be differentiated according to the age of presentation and evolution of the infection. In children and adolescents, granulomatous inflammation occurs mainly in the presinusoidal periportal spaces of the liver, leading, in most cases, to hepatosplenomegaly [20]. However, with no evidence of liver failure at this stage, the changes are largely reversible with treatment. Periportal fibrosis due to the deposition of collagen (also called Symmer’s tubular stalk fibrosis) occurs more frequently in chronically infected adults [21], leading to portal hypertension, portocaval shunting, ascites or esophageal varices, whose rupture may cause bleeding and death [2]. For the last three decades, abdominal ultrasound has become the best imaging technique for evaluating liver fibrosis caused by schistosomiasis [48]. Moreover, if available, magnetic resonance imaging provides a much greater contrast between the different soft tissue structures of the liver.

Monitoring treatment efficacy is difficult due to the absence of demonstrable eggs in stool or rectum biopsies, although would benefit from noninvasive imaging methods, allowing reliable fibrosis staging and estimation of vascular dysfunction. Measurement of anti-schistosomiasis antibodies is of limited use since most assays have positive results for at least 2 years or longer. Eosinophilia has limited sensitivity as an indicator of chronic schistosomiasis, and when normal, it cannot be used as an indicator of treatment success.

## 7. Neuroschistosomiasis

Neuroschistosomiasis can affect any region of the central nervous system (CNS), especially in the spinal cord and brain, and is seen in less than 5% of infected patients [49]. The prognosis depends largely on early treatment [50]. However, an autopsy study from Africa reported that half of the patients with urinary schistosomiasis had brain lesions [51]. For this to occur, the migration of adult larvae to the CNS and the deposition of eggs are necessary to initiate the chronic granulomatous inflammatory reaction. The manner in which the eggs reach the CNS may be through retrograde venous flow in the Batson vertebral epidural venous plexus or after embolization from portosystemic shunts or from left heart cavities. In addition, adult worms may migrate to settle in sites close to the CNS.

Some differences in morphology may explain differences in the clinical presentation between species. Thus, the smaller eggs of *S. japonicum* are more likely to be able to reach the central nervous system, whereas the larger eggs of other species (such as *S. mansoni* and *S. haematobium*) are more commonly found in the lower spinal cord [5]. The neurological complications of this parasitosis can occur in all phases of the disease; however, it is more frequent in the chronic phase, and the more severe the symptoms, the more intense the inflammatory reaction.

The onset of neurological symptoms usually takes place within weeks after infection and progresses in an acute or subacute manner, with the symptoms and signs of the disease progressively worsening. Approximately 90% of patients develop a full neurological picture within 2 months [52].

In cerebral schistosomiasis, the symptoms are caused by granulomatous lesions; by the deposition of eggs, accompanied by edema and mass effects distributed throughout the cerebral hemispheres; and by vasculitis and thrombosis-type lesions that may be induced by eosinophil toxicity [5]. Depending on the location, it will give a wide variety of symptoms: altered level of consciousness, signs of intracranial hypertension (headache, nausea and vomiting), focal neurological deficits or epileptic seizures, among others. The prevalence of epilepsy in communities where infections have occurred has been estimated at 1 to 4%, eight times as high as at baseline [33]. Occasionally, it may appear as a single brain granulomatous lesion that presents with partial seizures and may mimic an intracranial space-occupying lesion [35].

*Schistosoma mansoni* is the species that most frequently causes acute myelopathy. Acute transverse myelitis can cause flaccid and areflexic acute paraplegia, with sensory involvement and sphincter incontinence, whereas subacute myeloradiculopathy (Cauda equina syndrome) can present with weakness in the lower extremities, lumbar pain, saddle paresthesia, and vesico-intestinal incontinence.

Infection in childhood, especially with a history of malnutrition, can cause impaired cognitive development and learning disorders [36]. Clinical evidence of systemic disease is often absent in patients with neuroschistosomiasis, and the detection of parasite ova in urine and/or stool is possible in only 40% to 50% of cases [53]. Among the imaging diagnostics, Computed Tomography (CT) and MRI are of value in investigating neuroschistosomiasis, and several characteristic postcontrast MRI features in patients with cerebral schistosomiasis have been described. Cerebrospinal fluid serological testing can be helpful if the results are positive (sensitivity, 83–88%), but the test has relatively low specificity (range, 38–67%) [53].

## 8. Urogenital Schistosomiasis

### 8.1. Urinary Schistosomiasis

The species most frequently implicated at this level is *S. haematobium* [37]. This is because *S. haematobium*, unlike other human-infecting schistosome species, mainly migrates to the veins surrounding the bladder (the vesicle venous plexus), thus causing urinary schistosomiasis [38]. It is also remarkable the emergence of the concept of parasite hybridization, which is being observed more frequently in cases of urogenital schistosomiasis. The main impact is lower urinary, the site of heaviest oviposition. It typically affects the bladder, ureters, seminal vesicles, and, less commonly, the vas deferens and the prostate. Schistosoma lesions are typically limited to the lower halves of the ureters, corresponding to the lower border of the third lumbar vertebra [54]. The initial lesions are mucosal granulomas that coalesce to form tubercles, nodules or masses that usually ulcerate along the entire urinary tract. The surrounding mucosa is hyperemic. The submucosa and muscle layers are also involved in the inflammatory process, which may lead to transient back pressure if the ureterovesical junctions are affected. Involvement of the submucosa may lead to contraction of the bladder capacity. As they heal with excessive fibrosis, they may lead to strictures, calcifications and urodynamic abnormalities (Figure 5). Fibrosis of the muscle layer may contribute to bladder contraction and may also lead to urodynamic disorders, including an “irritable”, a “hypertonic” or an “atonic” bladder [54]. In this way, *S. haematobium* infection is classified as a Group I carcinogen because it can give rise to squamous cell carcinoma of the bladder, especially in endemic areas [5]. Secondary bacterial or viral infection is common, and any infection may be incriminated in secondary stone formation during the development of bladder malignancy, with rates from 0.5 to 25% [55]. The bladder outlet is one of the favorite sites for oviposition, being at the apex of the vesical trigone. Therefore, it induces a bladder neck obstruction.

The characteristic clinical presentation is terminal hematuria (Figure 6), usually associated with an increased frequency of micturition and dysuria, although it can also manifest as pyuria or hematospermia, the latter a common symptom in men [39,40]. In endemic areas, hematuria is the red flag of schistosomiasis in children aged 5 to 10 years. In the long term, they can cause chronic cystitis and ureterohydronephrosis, as well as papillomas and cancer.

### 8.2. Genital Schistosomiasis

Involvement includes any organ of the genital tract. Female genital schistosomiasis (FGS) is recognized as a common complication of *S. haematobium* parasitism, occurring in approximately half (33 to 75%) of infected females [56]. Thus, it is one of the most common gynecological conditions in endemic areas. FGS may also be a cofactor in the acquisition of HIV [57]. In this way, there was a three- to fourfold increased risk of acquiring HIV in women with a pre-existing *S. haematobium* infection. Female genital manifestations may include hypertrophic and ulcerative lesions of the vulva, vagina, and cervix; are associated with stress incontinence, decreased fertility, and abortions; and facilitate the sexual transmission of infections and an increased frequency of urination [58]. Highlighting the implications, both psychological and societal, of genital lesions in patients affected by urogenital schistosomiasis, particularly women. The largest study to date did not find any association between abortion or menstrual irregularities and *S. haematobium*. Genital self-sampling increased the overall number of PCR-based FGS diagnoses in a field setting, compared with cervicovaginal lavage. Home-based sampling may represent a scalable alternative method for FGS community-based diagnosis in endemic resource-limited settings [59]. Male genital manifestations may include involvement of the testicles, spermatic cord, epididymis, or prostate. Genital lesions may be partially reversible with treatment.

## 9. Pulmonary Schistosomiasis

Chronic Schistosoma infection can induce pulmonary arterial hypertension, a potentially fatal complication [60]. It is assumed that the development of pulmonary hypertension requires hepatosplenic schistosomiasis, which allows portal shunting and egg embolization [61]. The eggs can lodge in pulmonary arterioles and produce pulmonary hypertension and cor pulmonale. *S. mansoni* and *S. japonicum* are more frequently associated with the development of pulmonary hypertension, particularly *S. mansoni*. This form of chronic pulmonary disease causes dyspnea and cor pulmonale and may result in heart failure [5,62]. Radiological changes are nonspecific and can include ill-defined nodular, ground glass, or consolidative changes due to migratory parasites or egg deposition with focal granulomatous change. Both acute and chronic schistosomiasis can be successfully treated with praziquantel; however, improvement of the chronic manifestations, including pulmonary hypertension, has been less convincing [31].

## 10. Conclusions

Schistosomiasis is an important public health problem, particularly in poverty-stricken areas. The clinical manifestations are varied and affect practically all organs. There are substantial differences in the clinical presentation, depending on the phase and clinical form of schistosomiasis in which it occurs. Schistosomiasis can remain undiagnosed for a long period of time, with secondary clinical lesion. This information may aid in the development of more efficacious treatments and improved disease prognosis.

## Figures and Tables

**Figure 1 jcm-10-05521-f001:**
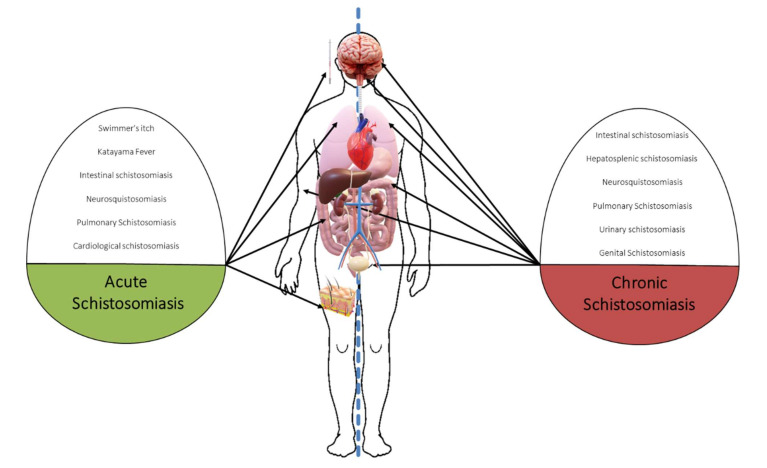
Main clinical manifestations of schistosomiasis (acute vs. chronic).

**Figure 2 jcm-10-05521-f002:**
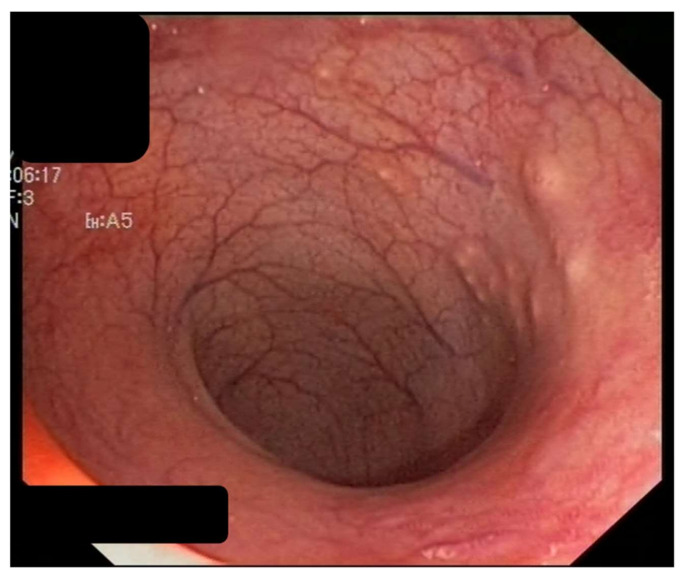
Intestinal schistosomiasis. Elevated yellow nodules and granular changes in colon suggestive of chronic colitis.

**Figure 3 jcm-10-05521-f003:**
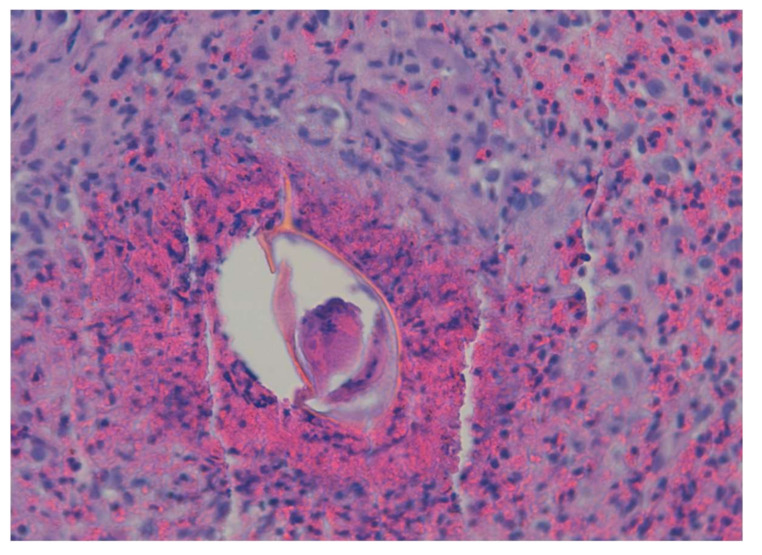
Eggs of *S. mansoni* present in the intestinal tract and granuloma (×40).

**Figure 4 jcm-10-05521-f004:**
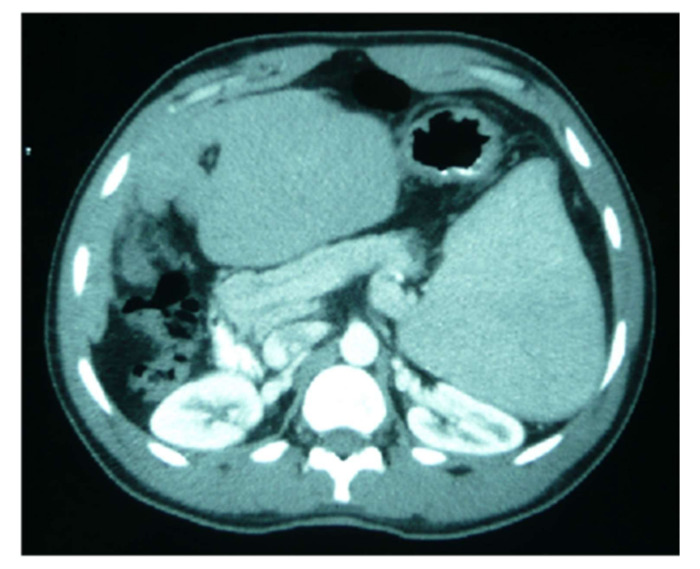
Splenic alargement due chronic schistosomiasis in a sub-Saharan patient.

**Figure 5 jcm-10-05521-f005:**
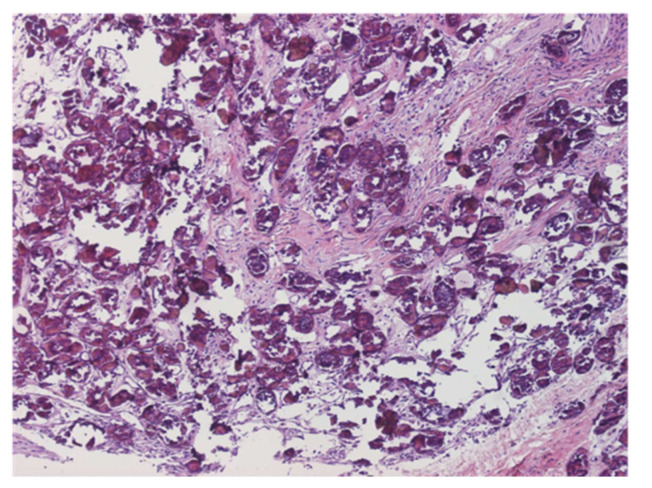
Fibrosis and calcifications in bladder tissue (×20).

**Figure 6 jcm-10-05521-f006:**
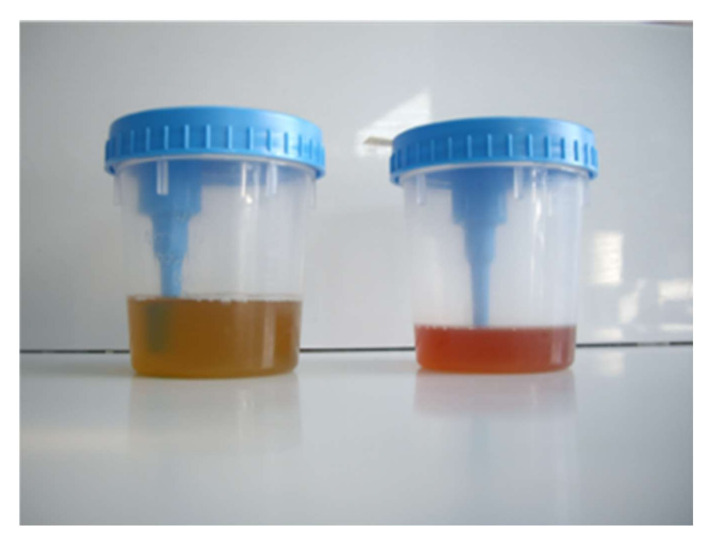
Terminal hematuria due *S. haematobium* in a young man from Mali.

**Table 1 jcm-10-05521-t001:** Classification of acute forms of schistosomiasis.

Types of Presentations	Clinical Manifestations
Swimmer’s itch	Local inflammation of the cercariae entry zone, most frequently caused by non-human pathogenic species that cannot migrate
Cercarial dermatitis	Maculopapular skin rash. It develops in previously sensitized people when they are reinfected by non-human pathogenic species
Katayama syndrome	Delayed systemic hypersensitivity reaction (3 and 8 weeks after exposure) It affects more than 50% of infected people. Fever, arthralgia, and cutaneous vasculitis and eosinophilia are the most common clinical manifestations. Spontaneous resolution after 2 to 10 weeks A minority develop persistent disease (weight loss, dyspnoea and diarrhoea, abdominal pain, hepatosplenomegaly)
Pulmonary form	Pulmonary symptoms resulting from the systemic immunoallergic reaction of acute schistosomiasis in non-immune patients. It presents as dyspnoea, bronchospasm, productive cough, haemoptysis, and/or chest pain, which may appear in isolation or within the clinical picture of Katayama fever

**Table 2 jcm-10-05521-t002:** Characteristics of schistosomiasis in residents in non-endemic areas and residents in endemic areas.

	Non-Endemic Area Resident	Endemic Area Resident
Most common form of disease	Acute schistosomiasis	Chronic infections
Age group	Adult	Children–adolescents–young adult
Most common clinical manifestations	Skin lesions (pruritus, skin eruption), fever, cough, abdominal pain, and diarrhoea	Anaemia, haematuria, abdominal pain, hepatomegaly
Diagnostic clues	History of exposure to fresh water in an area of endemicity	Abdominal pain, haematuria or/and genito-urinary symptoms More frequent ova identification and increased IgE

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
