# Peer review of "Clinical Spectrum of Schistosomiasis: An Update"

_jcm, 2021, doi:10.3390/jcm10235521_

Round 1
Reviewer 1 Report
In the "Acute schistosomiasis" section there is no information about species that cause this, which can be confusing (above information only about species occurring in humans).
"Some authors notes that using a single urine sample ..." - citation needed
Fig 2 description would be helpful
minor correction - example: Name S. haematobium - italics
Author Response
Response to Reviewer 1
- In the "Acute schistosomiasis" section there is no information about species that cause this, which can be confusing (above information only about species occurring in humans).
It is modified in the text
- "Some authors notes that using a single urine sample ..." - citation needed
Citation was included
- Fig 2 description would be helpful
Description was included
- minor correction - example: Name S. haematobium – italics
It is modified in the text

Reviewer 2 Report
This manuscript aims to review clinical aspects of urogenital and intestinal schistosomiasis. The subsections allow the reader to clearly focus on the pathology and symptomatology caused by schistosomiasis in different systems. Nevertheless, the manuscript misses to describe important aspects of the disease epidemiology, including parasite hybridization and its clinical aspects during human infection. Furthermore, the grammar of the text could be improved. I have listed some minor corrections below.
Title. Update instead of uptodate?
Abstract. Schistosoma in italics; please re-word “organs of the economy” (line 25) and “clinical injury” (line 27).
Lines 34-36. Missing reference.
Lines 36-38. The genus should be fully spelled out the first time a species is mentioned, to avoid confusion with other taxa.
Lines 40-44. The authors group schistosomiasis into three clinical stages but then further discuss only the acute and the chronic stages. Please add a sub-section about cercarial dermatitis including its epidemiology, parasites within the family Schistosomatidae that are involved, clinical profile, symptoms, and sequelae.
Line 47 and throughout the whole manuscript. The word “traveller” (please amend its spelling throughout the whole manuscript) is often repeated but I would rather highlight pathology in endemic areas. International travel and migratory movements can certainly be reported; however, the text seems often more focused on patients travelling/migrating from endemic regions rather than on the local communities that are living with the disease.
Lines 61-63. This statement should be more specific in describing the findings from refs. 16 and 17 and to which sub-section of the international travelling population these 3% and 5% refer to.
Lines 73-86. See above about adding an extra-section on cercarial dermatitis.
Lines 81-82. This is a generic statement that should be removed to keep the focus on the diagnostics of cercarial dermatitis.
Table 2. It is difficult to assign traits within the first column to which row they belong to; does the table need re-formatting? Also, this table is the first reference to “non-human pathogenic species that cannot migrate” and more details about these schistosome species should be added in the Cercarial Dermatitis sub-section.
Lines 93-95. Which term? Please specify.
Lines 111-113. Modify the text in these sentences since “darken the prognosis” and “ neurological clinic” do not seem correct.
Line 119. S. mansoni in italics.
Lines 144-145. Are these refs. specific to travellers and why not extending the clinical manifestations described to all affected subjects?
Lines 161-164. Any more recent estimates regarding the prevalence of urogenital schistosomiasis?
Lines 193-197. Missing refs. and further detail the importance of diagnostics in areas of low endemicity.
Lines 226-229. Would it be appropriate to quickly describe the standards of ultrasound examinations in the liver as in other organs to diagnose schistosomiasis (see also https://www.mdpi.com/2076-2607/9/8/1776).
Line 268. Avoid abbreviations of species names at the beginning of sentences.
Section 7 – Urinary Schistosomiasis. Urogenital schistosomiasis should be used as terminology since the disease involves both urinary tract and reproductive system. This section could be divided into two sub-sections describing urinary lesions and pathology in the reproductive organs. Furthermore, S. haematobium as causing agent should be reported alongside the emergence of the concept of parasite hybridization, which is being observed more frequently in cases of urogenital schistosomiasis (see also https://www.mdpi.com/2076-2607/9/8/1776, https://www.sciencedirect.com/science/article/pii/S2542519620301297).
Line 315. Schistosoma in italics.
Section 9 – Genital Schistosomiasis. See above and further develop this section by highlighting the implications, both psychological and societal, of genital lesions in patients affected by urogenital schistosomiasis, particularly women.
Conclusions. Should a section with concluding remarks be added to summarise the important points raised in this review?
Figures 2-6. Are these figures all unpublished material belonging to the authors? Ownership should be reported in the authors’ contributions if these figures are originals or permission to display them has been granted. The legend is extremely scant and it does not provide any detail for the reader; further description of each figure, and scale of the image where appropriate, should be reported.
Author Response
Response to Reviewer 2
This manuscript aims to review clinical aspects of urogenital and intestinal schistosomiasis. The subsections allow the reader to clearly focus on the pathology and symptomatology caused by schistosomiasis in different systems. Nevertheless, the manuscript misses to describe important aspects of the disease epidemiology, including parasite hybridization and its clinical aspects during human infection. Furthermore, the grammar of the text could be improved. I have listed some minor corrections below.
Thank you for the comments made. The aim of the work is focused on the clinical aspects of schistosomiasis not on the epidemiology, which is described very briefly.
- Update instead of uptodate?
It is modified in the text
- Schistosoma in italics; please re-word “organs of the economy” (line 25) and “clinical injury” (line 27).
It is modified in the text
- Lines 34-36. Missing reference.
Citation was included
- Lines 36-38. The genus should be fully spelled out the first time a species is mentioned, to avoid confusion with other taxa.
It is modified in the text
- Lines 40-44. The authors group schistosomiasis into three clinical stages but then further discuss only the acute and the chronic stages. Please add a sub-section about cercarial dermatitis including its epidemiology, parasites within the family Schistosomatidae that are involved, clinical profile, symptoms, and sequelae.
It is modified in the text
- Line 47 and throughout the whole manuscript. The word “traveller” (please amend its spelling throughout the whole manuscript) is often repeated but I would rather highlight pathology in endemic areas. International travel and migratory movements can certainly be reported; however, the text seems often more focused on patients travelling/migrating from endemic regions rather than on the local communities that are living with the disease.
It is modified in the text
- Lines 61-63. This statement should be more specific in describing the findings from refs. 16 and 17 and to which sub-section of the international travelling population these 3% and 5% refer to.
I think the phrase is clear. Thus, different studies show that more than 3% of international travelers have schistosomiasis, exceeding 5% of diagnoses if travelers came from sub-Saharan Africa.
- Lines 73-86. See above about adding an extra-section on cercarial dermatitis.
OK.
- Lines 81-82. This is a generic statement that should be removed to keep the focus on the diagnostics of cercarial dermatitis.
Please specify the sentence.
- Table 2. It is difficult to assign traits within the first column to which row they belong to; does the table need re-formatting? Also, this table is the first reference to “non-human pathogenic species that cannot migrate” and more details about these schistosome species should be added in the Cercarial Dermatitis sub-section.
It is modified in the table and in text
- Lines 93-95. Which term? Please specify.
It is modified in the text
- Lines 111-113. Modify the text in these sentences since “darken the prognosis” and “neurological clinic” do not seem correct.
It is modified in the text
- Line 119. S. mansoni in italics.
It is modified in the text.
- Lines 144-145. Are these refs. specific to travellers and why not extending the clinical manifestations described to all affected subjects?
These references are extended to all patients
- Lines 161-164. Any more recent estimates regarding the prevalence of urogenital schistosomiasis?
References are from 2010 and 2016.
- Lines 193-197. Missing refs. and further detail the importance of diagnostics in areas of low endemicity.
- Lines 226-229. Would it be appropriate to quickly describe the standards of ultrasound examinations in the liver as in other organs to diagnose schistosomiasis (see also https://www.mdpi.com/2076-2607/9/8/1776).
It is modified in the text
- Line 268. Avoid abbreviations of species names at the beginning of sentences.
It is modified in the text
- Section 7 – Urinary Schistosomiasis. Urogenital schistosomiasis should be used as terminology since the disease involves both urinary tract and reproductive system. This section could be divided into two sub-sections describing urinary lesions and pathology in the reproductive organs. Furthermore, S. haematobium as causing agent should be reported alongside the emergence of the concept of parasite hybridization, which is being observed more frequently in cases of urogenital schistosomiasis (see also https://www.mdpi.com/2076-2607/9/8/1776, https://www.sciencedirect.com/science/article/pii/S2542519620301297).
It is modified in the text
- Line 315. Schistosoma in italics.
It is modified in the text
- Section 9 – Genital Schistosomiasis. See above and further develop this section by highlighting the implications, both psychological and societal, of genital lesions in patients affected by urogenital schistosomiasis, particularly women.
It is modified in the text
- Should a section with concluding remarks be added to summarise the important points raised in this review?
It is modified in the text
- Figures 2-6. Are these figures all unpublished material belonging to the authors? Ownership should be reported in the authors’ contributions if these figures are originals or permission to display them has been granted. The legend is extremely scant and it does not provide any detail for the reader; further description of each figure, and scale of the image where appropriate, should be reported.
All figures are by the authors. It is modified in the text.
